# Telework during COVID-19: Effects on the Work–Family Relationship and Well-Being in a Quasi-Field Experiment

Maria José Chambel 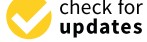, Vânia Sofia Carvalho and Alda Santos *

CICPSI, Faculdade de Psicologia, Universidade de Lisboa, Alameda da Universidade, 1649-013 Lisboa, Portugal
* Correspondence: aldasantos@psicologia.ulisboa.pt

**Abstract:** Due to the COVID-19 pandemic, organizations are forced to adopt teleworking. However, little is known about this work modality longitudinally. This study aims to clarify the impact of continuing to work on the organization's premises and shifting to a telework situation on the work and family relationship and employees' well-being. Using a sample of 435 bank employees with two waves, two groups were compared: (1) workers who continued to work on the organization's premises (213), and (2) workers' who had shifted to a telework situation (222). The first set of data were collected prior to the pandemic and the second approximately 10 months after its onset. The study found no statistically significant change to the work and family relationship (i.e., work–family conflict and work–family enrichment) as a result of a shift to telework. However, the shift to telework had a beneficial effect on work engagement, as opposed to remaining on the premises of the company. This study emphasizes the absence of effects on the work–family relationship resulting from the adoption of telework in response to COVID-19.

**Keywords:** work–family conflict; work–family enrichment; work well-being; general well-being; telework

## 1. Introduction

Within the scope of the COVID-19 pandemic and all the changes it has brought to people's lives, particularly in terms of the measures imposed on societies, the work–family relationship and employees' well-being emerge as issues that run deep [1]. Telework refers to an alternative work arrangement in which employees perform job-related tasks from a distant location (e.g., home) using electronic media [2]. It was one of the widely implemented measures used to contain COVID-19, with implications for the work–family relationship and employees' well-being [1,3,4]. In fact, as COVID-19 infections increased around the world, each country took steps to close down operations and, when possible, to introduce full-time telework for their employees. Telework, which was not widely implemented prior to the pandemic, and was generally viewed as a temporary solution, has now become part of the "new and better normal" and is likely to do so for years to come [4].

The imposition of telework during the pandemic, as a public health measure to prevent transmission of the virus, was rapidly adopted but did not allow for the preparation of workers or companies, some of whom were experiencing this work modality for the first time. Thus, this situation may be considered a crisis that is perceived and felt as a threat, given the associated difficulties that exceed one's habitual resources and coping mechanisms [5]. Therefore, an analysis of its consequences for the work–family relationship and the well-being of employees is of utmost relevance.

The aim of this research is to analyze the consequences of telework imposed as a COVID-19 containment measure for the work–family relationship and for employees' well-being. A sample of 435 bank employees was used in a two-wave study, the first wave prior to the pandemic and the second approximately 9 months after its onset. Research questions were advanced on the effects of telework on the work–family relationship (i.e., work–family

conflict and work–family enrichment), workplace well-being (i.e., burnout and work engagement), and general well-being (i.e., satisfaction with life and health perception).

This study offers several contributions to the literature on the work–family relationship and telework. Firstly, published studies contextualized in telework during the pandemic have shown some paradoxical results [6], which highlighted the pertinence of this study when analyzing the effects in the pre- and post-pandemic eras. By comparing the evolution of two groups, namely one consisting of the employees who adopted telework and the other of those who continued to work on the premises of the company, prior to and during the pandemic, this study contributes to clarifying the telework effect on employees' family–work relationship and well-being. Only by assessing the transition of these two distinct groups, prior to and after the pandemic, will it be possible to understand whether teleworkers have experienced a change for the better or worse [7,8]. Moreover, the literature on the work–family relationship [9] and that related to telework [10,11] has been mostly cross-sectional, which does not allow for an understanding of the changes over time. In this research, a longitudinal study was used with data collected at two points in time at a nine-month interval. Only longitudinal studies can contribute to making the approach to the work–family relationship more dynamic [12] and, especially, to improve our understanding of the consequences of telework [10,11].

Secondly, to our knowledge, studies on the effects of telework on the work–family relationship have tended to focus on their negative dimension, namely on whether they attenuate or aggravate its negative interference in terms of work–family conflict. Nevertheless, since telework is characterized by high levels of permeability and flexibility on the boundary of these domains, which may foster the transfer of resources (e.g., positive mood and attitudes) [13], this study adopts the original stance of analyzing the effect of this work modality not only on work–family conflict, but also on work–family enrichment [14].

Thirdly, most of the studies that have analyzed the effects of telework on well-being have only considered one dimension of well-being [10]. However, as suggested by Ryan and Deci [15] and Dodge, Daly, Huyton, and Sanders [16], well-being is a complex and multifaceted phenomenon that also encompasses optimal experience and functioning. Moreover, in line with the orientations of positive psychology, not only weakness and illness should be analyzed, but also strength and health. Thus, the aim of this study was not only to understand the effects of this work modality on ill-being (i.e., burnout), but also well-being (i.e., work, engagement, and satisfaction with life and health). Indeed, this study offers a broader perspective of the repercussions of telework, taking into consideration not only well-being at work (i.e., burnout and work engagement,) but also context-free well-being (i.e., general satisfaction with life and health) [17,18].

Finally, the literature related to telework has shown that the availability of and adherence to telework are highly dependent on the company's culture [19,20]. An analysis of the repercussions of this practice for a bank which, despite belonging to a traditionalist sector was forced to adopt telework in the context of the pandemic, may contribute to a better understanding of the repercussions of this solution and its adoption grounded on a future post-pandemic vision [3].

*Theoretical Foundation and Research Questions*

This pandemic emerged unexpectedly and expanded rapidly across the world. In Portugal, in line with the USA and many other European countries, a state of emergency was declared on 13 March 2020. As a result of this state of emergency, all companies across the country were forced to adopt telework and their workers were only permitted to continue performing their duties on the companies' premises when these duties could not be accomplished remotely.

In the case of the workers who shifted to telework, multiple positive and negative consequences for the work–family relationship may be expected, namely for the conflict between these two domains [21], defined as an inter-role conflict where the demands of the professional role are incompatible with the demands of the family role [22].

Several studies, including meta-analytical studies, have shown that telework has a beneficial, albeit slight, effect on work–family conflict [2,23]. In a situation of telework, there is greater flexibility and workers have more control over the pace and organization of their work and are able to regulate and coordinate the demands of both domains (e.g., they can interrupt work to prepare dinner or go to the supermarket for a last-minute purchase) [24,25]. This possibility of responding to the demands of family life during working hours facilitates the performance of the family role and contributes to a reduction in work–family conflict, as not only do teleworkers perceive that their working time does not prevent them from responding to family responsibilities, but they also feel less pressure since their work and family life are compatible [21,26].

However, other studies have shown that, on the contrary, telework is detrimental to the work–family relationship as it promotes conflict between these domains. When working at home, the physical boundary between the two domains is blurred [27] since the home is both the workplace and the place for family life [11], thus fostering border permeability which results in more distractions, interruptions, and violations of family life through work [28,29]. Conversely, this blurring of physical boundaries fosters a blurring of psychological and temporal boundaries, which translates into a blurred distinction between the two domains [30], coupled with an excess dedication to work, thus leading the teleworker to be constantly thinking about work and to perform professional tasks beyond the habitual working hours [31]. In fact, this tendency to prolong work after hours is due to the fact that the worker feels the need to either complete pending tasks or compensate for the time spent on domestic tasks during working hours [32].

These situations foster the emergence of work–family conflict, as not only do individuals spend a considerable part of their time carrying out professional tasks, leaving less time for the family domain, but also the stress caused by doing their work compromises the fulfillment of family-related responsibilities [21,26].

In line with the Conservation of Resources Theory [33], the excess dedication to work could represent a work role stress once it evokes a response to an environment in which there is the threat of a loss of resources, an actual loss in resources, or lack of an expected gain in resources [34]. Moreover, this situation of a loss of resources (e.g., time, energy, and attention) to perform work roles results that fewer resources being available to perform family roles, and high WFC is experienced.

It is also worth noting that when telework is imposed on workers and is not a choice, in line with the present case because of this pandemic, there is less control on the part of workers over how they manage the relationship between work and family. Even if they perceive that it would be more advantageous to accommodate the demands of these two domains by, for example, working some days at home and others at the office, they do not have this possibility [35]. This lack of volition also promotes work–family conflict [36].

On the other hand, teleworkers may have considered having the option of teleworking as a gain from a resource (e.g., flexibility and autonomy) that was previously not available to them. Thus, according to the basic principle of COR [33], teleworkers may have perceived telecommuting as a resource that allowed them to protect something they value, i.e., the family. However, Vitória and colleagues (2022) presented a systematic review of 25 studies that analyzed the effects of COVID-19 and observed that studies revealed WFC growth during the pandemic compared to before [37–39].

Based on these contradictions, we raise the following question.

RQ1: What are the effects of telework, adopted in the context of the COVID-19 pandemic, on work–family conflict?

Work–family enrichment refers to the extent to which experiences in one role improve the quality of life in the other [40] (p. 72); some characteristics of this situation make it possible to predict a positive effect. Work–family enrichment results from the acquisition of resources in the professional domain that are transferred to the family situation, contributing to the performance of the role in this latter domain [41]. As a teleworking situation involves the blurring of these domains, as a result of more permeable and flexible

physical, temporal, and psychological boundaries, it may foster this transfer of resources and promote work–family enrichment [13]. In fact, the ease of transition from the professional domain to the family domain that characterizes telework and which, as has been mentioned, promotes work–family conflict, may also be a channel through which the transfer of resources (e.g., positive mood and attitudes) of the performance of this role to the performance of the family role may be increased, namely work–family enrichment [14].

Moreover, the novelty of flexible work through telework might have been a resource that has positive effects on the work–family relationship [42] but also a perverted effect since roles easily overlap [43]. The study of Vaziri et al. [38] reports a profile of workers that have grown from low to medium scores of WFE but the other two profiles resulted in diminished work–family enrichment levels, from high to low scores or medium to low scores. Once more, based on these contradictories, findings we raise our second question.

RQ2: What are the effects of telework, adopted in the context of the COVID-19 pandemic, on work–family enrichment?

According to the COR, individuals strive to obtain, retain, foster, and protect those things they centrally value [33]. The theory posits that stress experienced by individuals can be understood in relation to potential or actual loss of resources. Accordingly, stress occurs (a) when central or key resources are threatened with loss, (b) when central or key resources are lost, or (c) when there is a failure to gain central or key resources following significant effort [44]. In this vein, burnout, defined as " . . . a state of exhaustion in which one is cynical about the value of one's occupation and doubtful of one's capacity to perform" [45] (p. 20), has been theorized as a state of resource depletion [46]. On the other hand, Schaufeli and Bakker [47] defined work engagement as a positive and fulfilling work-related state of mind characterized by vigor, dedication, and absorption. Engaged employees have a sense of energetic and affective connection with their work activities. The engagement literature identifies job resources as being crucial to arouse this positive state [47]. Building on the precepts of the COR theory, resource gain generates engagement. Furthermore, Hobfoll [33] defends that resource gain itself acquires salience in the context of resource loss and the literature already provides evidence of this proposition [48–50].

While teleworkers have more resources (e.g., they do not spend time commuting, they have greater flexibility to adjust their professional and family lives, greater control over work scheduling, and more autonomy to organize their tasks) then it may be assumed that they experience lower levels of stress [51,52] and higher levels of well-being [2,53]. In the same vein, some authors have stated that teleworkers experience fewer resource loss situations (e.g., frequent meetings and interruptions by colleagues [53]) than their colleagues who work on the premises of the company and, therefore, have less stress [54]. However, telework has also been associated with resource loss situations and, consequently, increased stress levels, especially in case of crises that evolve COVID-19 [4,6]. Teleworkers have been reported to work longer hours and put greater effort into the accomplishment of their tasks, thus leading to higher resource consumption (e.g., time and energy) and, consequently, to higher stress than their colleagues who are working at the office [30,55,56]. In the same vein, telework appears to favor the difficulty of re-establishing resources as the workers feel pressurized to be constantly available [57] and find it harder to switch off from work [20], thus jeopardizing their well-being and health [58]. On the other hand, teleworkers are more deprived of contact with their peers and supervisors, which may compromise social support, a fundamental resource for coping with stressful situations and ensuring the well-being of teleworkers [52,59]. Furthermore, recent studies focused on telework in lockdown have shown that those pandemic restrictions increased perceived stress and burnout [60–62].

Several studies have suggested that the emergence of burnout or engagement in workers is particularly important for an individual's general well-being and mental health [63–65]. Thus, the telework experience may also affect workers' satisfaction with life, which refers to the cognitive process by which people broadly assess the quality of their lives in various domains, e.g., family and work. Individuals who positively assess overall

satisfaction with life are satisfied with their life as a whole [66]. Moreover, this may also be reflected in health perceptions, which correspond to an overall assessment, not focused on specific health components (i.e., mental health, physical health, and physiological health). Health perceptions refer to individuals' explicit evaluation of their health, providing comprehensive information about the general state in which they evaluate their health based on objective information, as well as how they feel and evaluate this information [67]. Indeed, the study of Kumar and colleagues [62] that analyzed the impact of COVID-19 on stress observed that the life satisfaction of workers in India reduced due to a significant increase in distress levels and a decrease in job performance.

Therefore, the following research question was posited.

RQ3: What are the effects of telework, adopted in the context of the COVID-19 pandemic, on well-being in the workplace (i.e., burnout and engagement) and on well-being in a non-specific domain (i.e., satisfaction with life and perception of health)?

## 2. Method

### 2.1. Sample and Procedure

In February 2020, 1552 workers at a Portuguese bank were invited to participate in a study on the work–family relationship. A total of 1112 (71.6%) fulfilled the online survey during working time or at home. Anonymity was guaranteed and no incentive was proposed for their participation. In December 2020, all the participants were once more questioned to answer an identical questionnaire, and this time 1098 workers responded. At both points in time, the questionnaire was accessible for one month and the workers received two reminders during that period. Each participant created a code; it was possible to pair the responses of each participant at both points in time. A total of 435 valid questionnaires answered were obtained for the two waves of the study.

At the first point in time (T1), all the workers had been working on the premises of the bank. In March 2020, as a result of the country's lockdown due to the COVID-19 pandemic, all the bank workers with duties that could be performed remotely adopted telework and maintained this situation, even after the lockdown period had ended, until December 2020, the second point in time (T2). Thus, our sample consists of 435 participants who, at T1, had been working on the bank's premises and had no telework experience and, of whom, at T2, 213 (49%) continued working on the company's premises, while 222 (51%) had been teleworking for approximately 9 months.

A description of the sample in total and across groups is reported in Table 1. The majority of the sample (61.6%), was above 46 years old and the participants were mainly males (55.4%), married or in a stable relationship (74%), and with children (59.8%). Regarding the work-related variables, most of the participants were not supervisors/managers (75.2%) and had a job tenure of over ten years (89.2%).

**Table 1.** Sample Demographics.

| Sample | Total (*n* = 435) | Employees Who Remained in the Organization (*n* = 213) | Employees Who Shifted to Telework (*n* = 222) | χ2 |
|---|---|---|---|---|
| **Gender** (% males) | 55.4% | 59.6% | 51.4% | χ2 (*df* = 1) = 3, *n.s.* |
| **Age (Years)** | | | | χ2 (*df* = 4) = 10.67 * |
| Under 25 | 3% | 4.7% | 1.4% | |
| Between 26 and 35 | 6.7% | 7.5% | 5.9% | |
| Between 36 and 45 | 28.7% | 32.9% | 24.8% | |
| Between 46 and 55 | 50.8% | 46.5% | 55% | |
| Over 55 | 10.8% | 8.5% | 13.1% | |

**Table 1.** *Cont.*

| Sample | Total (*n* = 435) | Employees Who Remained in the Organization (*n* = 213) | Employees Who Shifted to Telework (*n* = 222) | χ2 |
|---|---|---|---|---|
| **Number of children** (less than 18 years) | | | | χ2 (*df* = 4) = 69, *n.s.* |
| 0 | 40.2% | 38.5% | 41.9% | |
| 1 | 29.9% | 30% | 29.7% | |
| 2 | 26.4% | 27.7% | 25.2% | |
| 3 | 3% | 3.3% | 2.7% | |
| 4 | 0.5% | 0.5% | 0.5% | |
| **Marital Status** | | | | χ2 (*df* = 3) = 1.31, *n.s.* |
| Single | 12.6% | 13.6% | 11.7% | |
| Married or living with a partner | 74% | 74.2% | 73.9% | |
| Divorced/Separated | 12.6% | 11.3% | 14% | |
| Widower | 0.7% | 0.9% | 0.5% | |
| **Job Position** (% not a supervisor) | 75.2% | 71.4% | 78.8% | |
| **Tenure (Years)** | | | | χ2 (*df* = 4) = 13.37 * |
| Under 5 | 5.3% | 4.7% | 5.9% | |
| Between 5 and 10 | 5.5% | 8% | 3.2% | |
| Between 10 and 20 | 38.2% | 41.3% | 35.1% | |
| Between 20 and 30 | 41.1% | 37.1% | 45% | |
| Over 30 | 9.9% | 8.9% | 10.8% | |

* $p > 0.05$; df = degrees of freedom; *n.s.* = non-significant.

As may be observed in Table 1, there were no significant differences between the workers in the two types of transition patterns (i.e., maintaining their work in the organization and shifting from the organization to telework) in terms of gender, marital status, number of children, and having a supervisor/manager position. However, there were differences in terms of age and job tenure. These differences appear to suggest that the workers who shifted to telework were older and had a higher job tenure.

*2.2. Measures*

*Work–family conflict.* We measured work–family conflict using nine items of the Portuguese version [68] of Carlson, Kacmar, and Williams' [69] scale. Example items included "After work, I am too tired when I come home to do some of the things I'd like to do" and "My job takes time from me that I would like to spend with my family/friends". The items were answered on a 5-point rating scale ranging from hardly ever (1) to almost always (5), and higher scores indicated greater interference of work in the family. Cronbach's alpha was 0.96 at time 1 for the whole sample and, at time 2, was 0.95 for workers who remained in the organization and 0.96 for workers who had shifted to a telework situation.

*Work–family enrichment.* Work–family enrichment was measured using nine items of the Portuguese version [68] of Carlson, Kacmar, Wayne, and Grzywacz's [70] scale. Two a sample items are as follows: "My involvement with my work helps me to understand different viewpoints and this helps me be a better family member" and "My involvement in my work helps me to develop my abilities and this helps me be a better family member". Items were scored on a 5-point rating scale from (1) totally disagree to (5) totally agree. At time 1, Cronbach's alpha was 0.92 for the whole sample and, at time 2, Cronbach's alpha was 0.92 for workers who remained in the organization and 0.93 for those who had shifted to telework.

*Work engagement.* Work engagement was measured with a Portuguese version [71] of a short version of the scale of Schaufeli, Bakker, and Salanova, [72] that included vigor (three items, e.g., "When I wake up in the morning, I feel good about going to work), dedication (three items, e.g., "My work inspires me;"), and absorption (three items, e.g., I am immersed

in my work). Respondents answered the items on a 7-point scale ranging from never (1) to every day (7). At time 1, Cronbach's alpha was 0.92 for the whole sample and, at time 2, was 0.92 for workers who remained in the organization and 0.95 for workers who had shifted to a telework situation.

*Burnout.* Burnout was assessed using a Portuguese version [71] of a scale [45] that included as core dimensions, exhaustion (five items, e.g., "I feel emotionally drained by my work") and cynicism (five items, e.g., "I question the significance of my work"). Respondents answered the items on a 7-point scale ranging from never (1) to every day (7). At time 1, Cronbach's alpha for cynicism and exhaustion was 0.85 and 0.93, respectively. At time 2, with regard to cynicism and exhaustion, Cronbach's alpha for workers who remained in the organization was 0.84 and 0.93, respectively, and 0.79 and 0.91 for those who had shifted to a telework situation.

*General well-being (context-free well-being).* Workers' general well-being was measured by means of their satisfaction with life and health perceptions. Satisfaction with life was assessed with the 5-item scale of Diener, Emmons, Larsen, and Griffin [73] that had been used previously in a Portuguese study [74]. Sample items include "I am satisfied with my life". The items were scored on a 7-point rating scale that ranged from totally disagree (1) to totally agree (7), with higher scores indicating a greater degree of satisfaction with life. The internal consistency reliability coefficient among participants in the present sample was 0.89 at time 1 for the whole sample 0.91 at time 2 for the workers who remained in the organization, and 0.90 for those who had shifted to a telework situation.

The Health Perceptions Questionnaire developed by Ware, Davies-Avery, and Donald [67] was used to assess health perceptions. This 4-item scale was used in a previous Portuguese study [74]. Sample items include "I am as healthy as others". The items were rated on a 5-point rating scale that ranged from absolutely false (1) to absolutely true (5). At time 1, Cronbach's alpha was 0.86 for the whole sample and, at time 2, Cronbach's alpha was 0.82 for workers who remained in the organization and 0.80 for those who had shifted to a telework situation.

*Control variables.* Gender, age, marital status, and number of children have been highlighted as variables that can affect the work–family relationship [75] and the effects of telework [76]. To control the effect of these variables, gender (dummy variable, 0 = female; 1 = male) and age (ordinal variable, 1 = up to 25 years; 2 = from 26 to 35 years; 3 = from 36 to 45 years; 4 = from 46 to 55 years; 5 = above 55 years) were considered. Adapting the strategy of Bedeian, Burke, and Moffett [77], a family demand variable was created, considering the marital status and the number of children (1 = single, divorced, or widowed without children; 2 = married without children; 3 = married, divorced, or widowed with 1 child; 4 = married, divorced, or widowed with 2 children; 5 = married, divorced, or widowed with 3 children; 6 = married, divorced, or widowed with 4 children), simultaneously.

### 2.3. Statistical Analyses

As a preliminary step of the statistical analyses, dropout analyses were carried out to examine the differences between the participants who had dropped out at time 2 (*n* = 677) versus those who had continued their participation at time 2 and constituted the present sample (*n* = 435) under study.

The fitness of the measurement models was first ascertained, following the recommendations of Pitts, Weis, and Tein [78]. The longitudinal measurement invariance was analyzed across the time points, the extent to which a certain construct remained the same over time and stability, i.e., the degree to which the relative ordering of subjects remains the same over time. Accordingly, a nested measurement model in which the factor loadings of all the relevant constructs were restricted to be equal across the two waves was compared with a measurement model in which these factor loadings were set free. The chi-square difference statistic was used to determine the extent to which this assumption held true. Mplus 7.4 [79], with the robust maximum likelihood estimator (MLR), was used. The fit indices employed were the comparative fit index (CFI), the root mean square error of

approximation (RMSEA), the standardized root mean square residual (SRMR), and the chi-square difference tests. For CFI, values greater than 0.90 represent a good model fit, and for SRMR and RMSEA, values below 0.07 indicate a good model fit [80]. Additionally, in order to control for the common method variance, our structural model was compared with a one-factor model (in which all items were loaded onto a single latent variable).

The hypotheses were tested using repeated measures ANCOVA, with time as the within-group variable (i.e., the repeated measure variable) and the transition pattern (work at organization-to-work at the organization and work at the organization-to-telework), as the between-group variable (i.e., the independent variable). Repeated measure designs, such as the one used in this study, reduce the error that might be due to individual variability within a population, a problem more commonly found in cross-sectional designs. This type of repeated measures design is used more frequently in longitudinal studies, in line with our purpose. Prior to performing these analyses, it was confirmed in all the groups that there were no significant outliers, that the dependent variable was approximately normally distributed, and that the variances of the differences between groups were equal. To assess change within each group and between groups, pairwise comparisons were inspected with Bonferroni correction.

## 3. Results

No differences were found in the dropout analyses. As no differences were found between those who had dropped out in comparison with those who continued participation in any of the variables, the response rate was considered acceptable.

As far as the measurement model is concerned, the fit indices for the measurement model (see Table 2) with free-factor loadings across the two points of time yielded acceptable results, $\chi^2$ (4146) = 7842.87, $p < 0.01$; CFI = 0.90; RMSEA = 0.04, SRMR = 0.07. A one-factor model was then computed ($\chi^2$ (4229) = 21,715.48, $p < 0.01$; CFI = 0.53; RMSEA = 0.10, SRMR = 0.11) and compared with the measurement model $\Delta\chi^2$ (83) = 13,872.61, $p < 0.01$). These results confirmed the construct validity of the measurement model. The model in which the factor loadings for each construct of the four variables were limited to be equal across the two points of time was then nested within the measurement model with free-factor loadings. The model with equal factor loadings yielded acceptable results ($\chi^2$ (4186) = 7895.98, CFI = 0.90; RMSEA = 0.04, SRMR = 0.07, $p < 0.01$) and differed significantly in fit, compared to the model with free-factor loadings ($\Delta\chi^2$ (40) = 53.11, $p < 0.01$). Thus, the model with equal factor loadings was preferred over the model with free-factor loadings, supporting the measurement invariability of the measured constructs.

**Table 2.** Testing construct validity and longitudinal constraints of the measurements' models.

| Model | $\chi^2$ | DF | CFI | RMSEA | SRMR |
|---|---|---|---|---|---|
| One-factor model | 21,715.48 ** | 4229 | 0.53 | 0.10 | 0.11 |
| Free factor loadings model | 7842.87 ** | 4146 | 0.90 | 0.04 | 0.07 |
| Equal factor loadings model | 7895.98 ** | 4186 | 0.90 | 0.04 | 0.07 |

** $p < 0.01$.

As regards the work–family relationship (i.e., work–family conflict and work–family enrichment, Table 3), there was no significant interaction effect. However, when considering work–family conflict, the results suggested that the workers who had shifted to telework were more inclined to report lower work–family conflict as the main effects observed were related to group and time [$F$ (1, 435) = 4.17, $p < 0.05$; $F$ (1, 435) = 7.44, $p < 0.01$; respectively]. Additionally, for work–family enrichment, a significant main effect of time [$F$ (1, 435) = 10.10, $p < 0.01$] was also observed, thus suggesting that both the workers who had shifted to telework and those that had remained on the premises tended to change their perceptions of work–family enrichment over time similarly. Therefore, telework, adopted in the context of the COVID-19 pandemic, did not change the work–family relationship in terms of conflict or enrichment.

**Table 3.** Longitudinal results for employees that remain in the organization and employees who shifted to telework.

| | Employees Who Remained in the Organization ($n$ = 213) | | | | Employees Who Shifted to Telework ($n$ = 222) | | | | Groups (between Groups) | | Time (within Groups) | | Group × Time | |
|---|---|---|---|---|---|---|---|---|---|---|---|---|---|---|
| | Time 1 | | Time 2 | | Time 1 | | Time 2 | | $F$ | $p$ | $F$ | $p$ | $F$ | $p$ |
| | Mean | SD | Mean | SD | Mean | SD | Mean | SD | | | | | | |
| **Work–Family Relationship** | | | | | | | | | | | | | | |
| Work–family Conflict | 3.11 | 0.99 | 2.98 | 0.96 | 2.70 | 0.93 | 2.54 | 0.87 | 4.17 | 0.05 | 7.44 | 0.01 | 1.15 | 0.26 |
| Work–family Enrichment | 2.94 | 0.75 | 2.98 | 0.77 | 3.10 | 0.76 | 3.20 | 0.76 | 3.58 | 0.06 | 10.10 | 0.01 | 0.78 | 0.78 |
| **Workplace Well-being** | | | | | | | | | | | | | | |
| Burnout | 3.45 | 1.38 | 3.47 | 1.38 | 3.14 | 1.33 | 2.98 | 1.18 | 1.17 | 0.28 | 7.08 | 0.01 | 0.82 | 0.80 |
| Engagement | 5.09 | 1.26 | 5.21 | 1.30 | 5.28 | 1.21 | 5.52 | 1.06 | 2.86 | 0.10 | 3.50 | 0.00 | 1.82 | 0.01 |
| **General Well-Being** | | | | | | | | | | | | | | |
| Satisfaction with Life | 4.25 | 1.23 | 4.39 | 1.32 | 4.57 | 1.26 | 4.64 | 1.22 | 2.59 | 0.11 | 8.15 | 0.01 | 1.29 | 0.16 |
| Health Perceptions | 3.07 | 0.88 | 3.04 | 0.82 | 3.28 | 0.77 | 3.30 | 0.74 | 1.14 | 0.29 | 11.46 | 0.00 | 1.27 | 0.22 |

$F$ values refer to the main and interaction effects of group and time. Gender, age, and family demands were controlled.

For workplace well-being (i.e., burnout and work engagement) and general well-being (i.e., satisfaction with life and health perception), there was only a significant interaction effect of work engagement [$F$ (1, 435) = 1.82, $p < 0.01$]. More specifically, the workers who had shifted to telework reported an increase in work engagement. On the other hand, those who had continued to work at the organization maintained their levels of work engagement. In addition to these interaction effects, the main effects of time were also found for burnout, satisfaction with life, and health perception [$F$ (1, 435) = 7.08, $p < 0.01$, $F$ (1, 435) = 8.15, $p < 0.01$ and $F$ (1, 435) = 11.46, $p < 0.01$, respectively]. However, since the Group × Time interaction effect was not significant, both the workers who had shifted to telework and those who had continued to work at the organization tended to change their perceptions of burnout, satisfaction with life, and health perception over time similarly. Therefore, telework adopted in the context of the COVID-19 pandemic increased work engagement but did not alter burnout, satisfaction with life, or health perception.

## 4. Discussion

The results of this study refute the idea that telework, in contrast to the situation of continuing to work on the premises of the company, may bring advantages or disadvantages to the work–family relationship. When workers were removed from the organization to perform their work tasks remotely, neither the conflict nor enrichment of the work–family relationship was affected, which was not the case with the workers who continued to work at the company. This result is the same as that observed in the quasi-experimental study of Delanoeijen and Verbruggen [81], in which the effects of implementing telework were compared at least twice a week with workers' permanence on the company's premises, and significant differences were not found in the work–family conflict. It is possible that the beneficial effect of telework on the work–family relationship may have been absent as this effect depends on teleworkers' ability to use the flexibility provided by this type of work, which may not occur if employees, due to lack of experience, are not able to use this flexibility [82]. In the current study, this justification is quite relevant as the bank from which the sample was collected had never adopted telework prior to the pandemic and, therefore, the workers in our sample had no previous experience with this modality of work. In addition, since telework was implemented to respond to an emergency situation, there was no time for workers or the company to prepare for its implementation. However, having observed that there were no harmful effects for the teleworkers, it may be assumed that they were able to manage their professional and family domains. This result contradicts that of Kaduk, Genadek, Kelly, and Moen [83], who observed that IT workers in an imposed telework situation evidenced more work–family conflict. However, this may perhaps be explained by the results obtained by LaPierre et al. [35], in which a detrimental effect of telework imposed on the work–family conflict only occurred when workers had weak

work–family balance self-efficacy, since when it was strong, telework no longer had a significant effect on this negative interference of work in the family. In the current study, teleworkers did not consider themselves to have high work–family conflict, and it may be assumed that they would have considered themselves to be relatively effective in establishing a balanced relationship between these two domains of their life [2,23].

Another explanation for this result may be the increase in organizational, co-worker, and supervisor support in the context of pandemic crises that may mitigate the possible detrimental effects of imposed telework on the WFC [39,60,84]. Additionally, this study collected data in two waves with a 10-month time lag. As [33] argues, the stressors that result in attrition can be temporary and, if they are not chronic, they do not evolve into a loss spiral. Thus, as our results report during the adaptation of telework, workers may have a WFC but 10 months later they may have found resources to combat this stressful situation.

Our results suggest that the shift to telework had a beneficial effect on well-being at work as opposed to remaining at the company. Those who adopted telework had an identical evolution to those who remained at the company in terms of their levels of burnout, satisfaction with life, and health perception. However, their levels of work engagement increased, while this positive psychological state of well-being at work did not change for the workers who were kept at the company. Indeed, this result may indicate that despite teleworkers' perception that this shift to telework involves having to dedicate more time to their work, leading to energy consumption, i.e., resources, it may also suggest that they interpret this shift as an opportunity to have more flexibility in scheduling their work and more control over their tasks, that is, having different resource gains which serve as a source of motivation in their work.

On the other hand, the fact that no changes to health perceptions and satisfaction with life were observed may be related to these variables not being solely dependent on individuals' work situation, but also their subjective assessment of life as a whole. Thus, all the workers, regardless of whether they remained at the organization or shifted to telework, live in the same social context, namely the pandemic context of COVID-19. Indeed, studies report COVID-19 has a great impact on the population due to economic crises that can be considered as loss-related events and can cause a lack of group resources, or can cause threat with loss, or can cause a failure to gain [85]. However, all workers in our sample maintain their employment and have high employment security, which involves the maintenance of resources, and their general well-being was not affected.

### 4.1. Theoretical Implications

Regarding the work and family relationship, as reported, our study did not find significant differences between the group of teleworkers compared with the group that remain to work in the company. This result may be interpreted in line with the basic COR theory tenet, that is, individuals strive to obtain, retain, foster, and protect things they centrally value, i.e., the family [33]. In this fashion, teleworkers may develop strategies to adapt to telework during the 9 months of the data collection interval. Moreover, with the idea in mind of protecting their family, teleworkers may be more aware of resources capable of protecting them from the possible spillover of negative effects for the family due to telecommuting.

### 4.2. Practical Considerations

From a practical point of view, the absence of these negative results may contribute to a change in managers' negative view of telework, which may serve as the "culture shock" required to facilitate long-term cultural changes in relation to remote work. This new perspective is essential to ensure that in the post-COVID period, telework may be accepted and adopted by organizations and their workers [4]. Thus, it is of the utmost importance that, by giving workers the possibility of having the flexibility of working from home, they provide them with contextual resources in order to facilitate this change, as well as promoting support from management and colleagues [39,60,84]. Moreover, it is

relevant that workers also develop strategies to adapt to telework, such as, for example, segmentation strategies that allow them not to extend working hours [86]. Finally, companies must consider the feelings of injustice that may arise, given that this possibility of teleworking may only be possible for some workers, finding a way to compensate others for not having access to this measure of flexibility.

### 4.3. Final Considerations

In the observed results, the absence of effects resulting from the adoption of telework in response to COVID-19 in the work–family relationship is noteworthy. The absence of positive effects may have occurred because telework was unexpectedly adopted without time for preparation by the workers and their supervisors [87]. However, the absence of negative effects is surprising, as telework was an imposed measure and not a voluntary option for the workers [83]. This created some degree of inequality within the company where some workers remained on the premises while others shifted to telework [88]. Perhaps the teleworkers felt privileged as their protection against the possibility of becoming infected was boosted, thus offsetting the possible negative effects. Additionally, it may be considered that as approximately 9 months had elapsed since the implementation of telework, the workers had already had time to develop the adaptive strategies that enabled them to dispel the short-term negative effects of a crisis situation [89].

In fact, the forced experience of telework during lockdown may have given rise to extremely difficult situations for these workers in the management of their work–family relationship, since in many cases several members of the same household were working from home, including children and adolescents/young adults in a distance learning format. However, in the later phase of the study, these workers were already in a more stable situation (e.g., the children and adolescents/young adults had resumed face-to-face education) and would have had time to develop adaptive strategies for this situation (e.g., defining a specific place and time to work at home) that allowed them to re-establish the work–family relationship, thus obtaining better levels of conflict than prior to the pandemic, when they had been working on the premises of the organization [2]. In addition, it may be assumed that the organization itself will have had time to develop monitoring and support that may have also promoted this adaptation to telework [5].

### 4.4. Strengths, Limitations and Future Studies

The major strengths of the current study are its longitudinal two-wave design and its focus on the transition to telework as a consequence of COVID-19 in relation to workers' work–family relationship and well-being. Moreover, to our knowledge, it is the first study to analyze this transition because of the pandemic, with a focus on two trajectories: from work on the organization's premises to telework and continuing to work at the organization. Not only the negative (work–family conflict) and positive (work–family enrichment) indicators of the work–family relationship are analyzed, but also indicators of well-being at work, including work engagement and burnout, and indicators of general well-being, namely health perception and satisfaction with life. Despite these strengths, the study also presents some important limitations.

The main limitation is that the analysis of a single company leads to a low external validity of the results. Thus, caution should be taken in generalizing these results to the consequences of telework due to the COVID-19 pandemic in other companies, since our results may also be related to the characteristics of this particular company (e.g., organizational culture, leadership style, and HRM practices), which may have influenced the manner by which this type of work was adopted and, thereby, its consequences [87].

Secondly, in a pandemic situation, continued commuting to work increases the risk of infection, which may have caused the group of workers who continued to work on the premises of the company to consider this decision to be unfair, despite being based on the nature of their role in the company. As the perception of justice can affect workers' reaction

to the implementation of telework [88], the results of this study in the group of workers who remained at the company may have been affected. By the same token, the workers who shifted to telework may have tended to respond favorably to the questionnaire as a means of "rewarding" the organization for its initiative. Furthermore, as some significant differences between the two groups were observed prior to the pandemic, which were generally more favorable for those who had switched to telework, the absence of effects or the positive effects observed in the workers who shifted to telework may not have occurred in the workers who remained at the company, even if they had switched to telework.

Thirdly, different individual variables (e.g., work–family border management, [21,38]) and the context (e.g., family-supportive organizational perceptions (FSOP), and supervisor work–family support [90]) are crucial to explaining workers' adaptation to telework and the repercussions of this modality for the work–family relationship. Therefore, their non-inclusion in this study is a shortcoming. Additionally, some diary studies have shown that telework can lead to within-person change [21,81]. Thus, the development of diary studies would be important in the future, including variables related to the border management of individuals and the context in order to assess the repercussions of telework in addition to a cumulative "mean" group change.

Other explanations for these results may also be advanced and could be explored in future studies. For example, it is well established that both the variables related to the work–family relationship (conflict [22] and enrichment [40]) and those related to well-being (well-being at work, burnout and engagement, and subjective well-being) are dependent on other variables, such as job characteristics [47]. Furthermore, these job characteristics may take on different importance when the workers are in telework [52]. Such examples may be observed in the study of Golden, Veiga, and Simsek [91], which showed that telework reduces the WFC and was even more pronounced for employees reporting higher levels of job autonomy and scheduling flexibility, which presumably allowed them to arrange their work tasks in such a way as to accommodate their family or other non-work commitments. Moreover, the study of Sardeshmukh et al. [52] revealed that job demands and resources mediated the relationship between the amount of time spent teleworking, exhaustion, and engagement, again suggesting the impact of work characteristics on the outcomes of the teleworkers. Additionally, some studies have shown that well-being at work is also dependent on how workers perceive the relationship between work and their personal life [92,93]. Thus, future studies should explore the relations among these variables, particularly in the context of telework. On the other hand, both job characteristics and individual variables as coping mechanisms have been deemed crucial for regulating states of well-being [46]. Future studies, therefore, should investigate whether individual strategies, such as coping, are effective for well-being in a telework situation. Finally, an assessment of the impact of telework on different individuals' demographic characteristics (for example, age, sex, and marital status) should also be explored in future research.

**Author Contributions:** Data curation, M.J.C.; Formal analysis, M.J.C. and V.S.C.; Funding acquisition, A.S., M.J.C. and V.S.C.; Methodology, M.J.C.; Writing—original draft, M.J.C.; Writing—review and editing, M.J.C., V.S.C. and A.S. All authors have read and agreed to the published version of the manuscript.

**Funding:** This work was financially supported by FCT—Fundação para a Ciência e a Tecnologia, I.P, granted to the project "Work–Family boundary dynamics in nontraditional jobs" (PTDC/PSI-GER/32367/2017). This work also received national funding from FCT—Fundação para a Ciência e a Tecnologia, I.P., through the Research Center for Psychological Science of the Faculty of Psychology, the University of Lisbon (UIDB/04527/2020; UIDP/04527/2020).

**Institutional Review Board Statement:** The study was conducted according to the guidelines of the Declaration of Helsinki and approved by the Ethics Committee of the Faculty of Psychology, the University of Lisbon.

**Informed Consent Statement:** All participants in the study gave their informed consent prior to their participation.

**Data Availability Statement:** The data presented in this study are available upon request from the corresponding author.

**Conflicts of Interest:** The authors declare no conflict of interest.

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
