# Peer review of "Telework during COVID-19: Effects on the Work–Family Relationship and Well-Being in a Quasi-Field Experiment"

_sustainability, doi:10.3390/su142416462_

Round 1
Reviewer 1 Report
Dear authors,
Thank you for the opportunity of reading the article. The results obtained are interesting as they are in accord to previous pandemic research on flexible arrangements, but are in contradiction with more recent studies that underline the negative impact of teleworking on work-life balance increasing work-life conflicts.
Methodology is clearly presented.
Theoretical contributions are well underlined. Could authors provide more insight on COR implications on work-family conflict? Also discussions should be more developed including comparisons with previous research in COVID-19 context.
Conclusions should be structured in theoretical contributions to COR theory and teleworking theory development, practical implications and then limitations respectively future directions of the research.
Reviewer 2 Report
1- Most of the sample was at the age of 45, and I think we needed to highlight the differences between different ages
2- I highly respect the researchers’ sincerity in confirming that there are no differences between workers at work and those away from work, but it remains that there are many reasons for the absence of differences, including the nature of work, age of the sample, fear of injury...etc.
So we need a deeper discussion
Reviewer 3 Report
Comments to the Author(s)
Recommendation: Major revision
First, I would like to thank the Editor for trusting me with the opportunity to review this research, "Telework during COVID-19: effects on work-family relationship and well-being in a quasi-field experiment.” The manuscript addresses the timely and prevalent topic of telecommuting and its impact on employees’ work-family conflict, work-family enrichment, work engagement, burnout, and general well-being using a pre-post design. The manuscript seems to have potential to contribute the extant literature on telecommuting subject to a major revision. Below are some suggestions that need to be taken care of.
1- The introduction seems fine and well-articulated to a greater extent. However, the authors have merged the introduction section with the literature and the study also lacks most recent references from the field of telecommuting research. I suggest the authors to separate the theoretical foundation from the introduction section and let the research questions part of introduction section. In addition, the research question is a bit too general and does not reflect what problem the study is exactly addressing as it talks about only the influence of telework on work-family conflict. I suggest the authors to come up with more specific research questions addressing all the variables that are included in the study. Some suggested references are given below.
Jamal, M.T., Anwar, I. and Khan, N.A. (2022), "Voluntary part-time and mandatory full-time telecommuting: a comparative longitudinal analysis of the impact of managerial, work and individual characteristics on job performance", International Journal of Manpower, 43(6), pp. 1316-1337. https://doi.org/10.1108/IJM-05-2021-0281
Hayes, S. W., Priestley, J. L., Moore, B. A., & Ray, H. E. (2021). Perceived Stress, Work-Related Burnout, and Working From Home Before and During COVID-19: An Examination of Workers in the United States. SAGE Open, 11(4), 21582440211058190.
Jamal, M. T., Alalyani, W. R., Thoudam, P., Anwar, I., & Bino, E. (2021). Telecommuting during COVID 19: A Moderated-Mediation Approach Linking Job Resources to Job Satisfaction. Sustainability, 13(20), 11449. https://doi.org/10.3390/su132011449
Kumar, P., Kumar, N., Aggarwal, P., & Yeap, J. A. L. (2021). Working in lockdown: The relationship between COVID-19 induced work stressors, job performance, distress, and life satisfaction. Current Psychology, 40(12), 6308–6323.
2- The study lacks the literature and hypotheses development section. Given that the study attempts to look for changes in study’s variables, viz., work-family conflict, work-family enrichment, work engagement, burnout, and general well-being among the employees working from office and employees switched to telework across pre and post-pandemic times; thus, the author should add a literature and hypotheses section while proposing the relevant hypotheses.
3- Methods and results section seems fine.
4- The discussion section seems a bit general and lacks specificity. I suggest the authors discuss the findings more specifically while being critical of the findings of previous studies.
5- I also suggest the authors to add a separate section for the theoretical and practical implications of the findings.
Round 2
Reviewer 3 Report
I commend the authors for improving the manuscript by addressing all the suggestions. However, I suggest the authors make the "Strengths, limitations and future studies" section the last one.
Author Response
We thank the Reviewer for the suggestion and have altered the order of the sections accordingly, finishing the manuscript with the "Strengths, limitations and future studies" section. Also, we have spell checked our manuscript one more time.